# Transfer Learning Strategy in Neural Network Application for Underwater Visible Light Communication System

**DOI:** 10.3390/s22249969

**Published:** 2022-12-17

**Authors:** Zengyi Xu, Jianyang Shi, Wenqing Niu, Guojin Qin, Ruizhe Jin, Zhixue He, Nan Chi

**Affiliations:** 1Key Laboratory for Information Science of Electromagnetic Waves (MoE), Fudan University, Shanghai 200433, China; 2Pengcheng Laboratory, Shenzhen 518055, China

**Keywords:** UVLC, nonlinearity, DNN, computational complexity, generalization

## Abstract

Post-equalization using neural network (NN) is a promising technique that models and offsets the nonlinear distortion in visible light communication (VLC) channels, which is recognized as an essential component in the incoming 6G era. NN post-equalizer is good at modeling complex channel effects without previously knowing the law of physics during the transmission. However, the trained NN might be weak in generalization, and thus consumes considerable computation in retraining new models for different channel conditions. In this paper, we studied transfer learning strategy, growing DNN models from a well-trained ‘stem model’ instead of exhaustively training multiple models from randomly initialized states. It extracts the main feature of the channel first whose signal power balances the signal-to-noise ratio and the nonlinearity, and later focuses on the detailed difference in other channel conditions. Compared with the exhaustive training strategy, stem-originated DNN models achieve 64% of the working range with five times the training efficiency at most or more than 95% of the working range with 150% higher efficiency. This finding is beneficial to improving the feasibility of DNN application in real-world UVLC systems.

## 1. Introduction

It is predicted that 6G communication will include a visible light channel (400–700 nm) as its indispensable part, benefiting from the ultrahigh communication capacity, no patent limitation, environmental friendliness, and other advantages over conventional RF spectrum [1]. The transmission window located at 400–500 nm [2] enables building the VLC link within a decent distance (1~100 m). Compared with conventional acoustic or RF methods, underwater VLC (UVLC, or Underwater Optical Wireless Communication, UOWC) has an unparalleled advantage in channel capacity, low cost, and free from electromagnetic interference (EMI) [3]. It is envisioned in future underwater systems such as underwater IoT [4,5,6], and massive data exchange between wireless/cable interfaces [1,7,8]. There is already research demonstrating the potential [9,10,11,12] of UVLC adopting currently available coherent/incoherent visible light sources. These studies declare a new era for communication at gigabits under the surface of the ocean. However, compared with the promising perspectives, in the current stage, UVLC still faces challenges that limit its application. LED-based VLC systems currently cannot utilize the progress in coherent detection such as the quantum detection with coherent-state signals [13] but have to apply the direct-modulation direct-detection method (IMDD) in producing and receiving the optical signal, due to its non-spontaneous radiation. Intensive fluctuation in electrical-to-optical and optical-to-electrical conversion involves heavy nonlinearity. Therefore, once the signal power is large enough to cause more nonlinearity than the SNR improvement, its negative side would dominate and stops the increment of signal quality. 

A heated topic in underwater VLC is the method of compensating the nonlinearity throughout the whole communication process. Some research pays attention to the stage before transmission (pre-equalization) [14,15,16] while others focus on the stage at the receiver side (post-equalization) [17,18,19]. Pre-equalization can modify the waveform before it is sent to the channel. It includes hardware solutions [14,15] and software ones [16]. However, pre-equalization is limited by the power available for transmission. The attempt that tries to allocate power and achieves a flat power spectrum would cause intolerable power loss when the bandwidth is large [16]. Post-equalization processes the signal after the channel and hence avoids problems with signal power availability. 

Machine learning (or more often deep learning) is a powerful fitting tool to solve high-order nonlinearity and there are already studies on this idea [20,21,22]. On the one hand, it is more flexible to set the scale of the NN model by tuning its width and depth, and whether to apply a convolution layer in the model. Meanwhile, NN is versatile to complete multiple steps in traditional digital signal processing (DSP). On the other hand, the computation involved in training NN models is a considerable load for communication systems to bear. In real UVLC communication systems, the channel condition is complex and rapidly changes by flow, pressure, temperature, and salinity, and meanwhile, the signal power requirement in UVLC is usually high (also means high nonlinearity in light source and photodetector) due to the strong absorption and scattering [23]. Therefore, multiple models are needed to cope with the fast-changing and highly nonlinear channel.

There are existing studies focused on reducing the computational complexity in training NN. For instance, using LSTM or more simpler GRU model to reduce the complexity in post-equalizer [24,25,26]. Another approach is partitioning the signal by its temporal amplitude (PWL) [27], demodulated symbol (PCVNN) [28], or spectrum (TTHnet) [29] and training dedicated and simpler models. Although they indeed achieve a reduction in computational complexity, these methods still belong to the set of supervised learning branches in machine learning. Therefore, the sample sizes and computation costs in training a new model are the same as the old ones and challenge the data processing ability. 

However, another branch of ML: transfer learning (TL) could solve this problem. As the feature spaces and label spaces for those NN-based post-equalizers are the same, we could apply a homogeneous transfer learning technique to reduce the computational complexity [30]. The computation saved by TL also reduces energy consumption and shortens the interval between training new models offline, which facilitates online model updating [31]. TL categorizes the samples into two sets: source domain and task domain. The source domain is used to produce a well-trained model that could generally solve the problem, and the task domain is to finely tune the model parameter to let it finish specific tasks more accurately [30]. 

To support a high transmission rate in the VLC channel, common low-order modulation schemes are not practical, such as OOK and QAM. Some researchers prefer the OFDM/DMT modulation scheme [20,32,33,34], while some apply Carrierless Amplitude and Phase (CAP) modulation [35,36,37,38]. This experiment takes CAP as the modulation scheme. In complexity, CAP saves the hardware and software resources for FFT/IFFT and the modulation process is similar to common FIR filters. In those experiments, their transmission rates could easily achieve 1 Gbps. Some even exceed the milestone of 10 Gbps. This data rate is far beyond the reach of underwater acoustic and RF communication. 

In this article, the traditional training method that trains every model from a randomly initialized model is named the ‘exhaustive training strategy’. This research compares the traditional LMS/Volterra (second order), exhaustive training strategy, and transfer learning strategy in system performance and computational cost, and gives evidence to support the application of the proposed idea.

## 2. Materials and Methods

### 2.1. Experiment Setup

This experiment uses a 1.2 m water tank filled with tap water to represent the underwater environment. An AWG710B arbitrary waveform generator (AWG) by Tektronix, Beaverton, USA, is used to transmit the waveform samples at 2 Gsa/s. The signal is amplified using a ZHL-2-8-S+ amplifier and biased using a ZFBT-4R2FW-FT+ Bias Tee (both from Mini-Circuits, New York, USA) to drive the LED set. To detect the signal, we use a DSO9404A oscilloscope (OSC) by KEYSIGHT, Santa Rosa, USA, which can operate at 2 Gsa/s, too. As the symbol is 4x up-sampled, the actual bandwidth is 500 MHz. Therefore, the sampling rate of OSC is enough for the signal as its Nyquist frequency is beyond the signal bandwidth. In the context of 64-ary modulation, it means a 3 Gbps data rate.

This experiment adopts 64 QAM and 64 APSK modulation schemes to produce the complex-valued symbols. There were 51,200 symbols generated by pseudo-random code using MATLAB that constitute the transmitted data. The symbol is later converted by CAP modulation (see the method part). To train a model, the sequency is divided into a training set and validation set (also the test set) by 1:4. Traditional online/offline CAP modulation/demodulation and LMS/2nd Volterra post-equalization process is performed using MATLAB script on R2021b version. For DNN training, we use Python 3.9.12 on Spyder 5.2.2. The main module used is torch 1.10.2 version.

### 2.2. Carrierless Amplitude and Phase (CAP) Modulation

In Figure 1 the principle of CAP modulation is shown. Its feature is to merge two independent channels into the same waveform samples using a pair of orthogonal filters. In this experiment, we first code the original data into a complex-valued using the mapping table of 64 QAM or 64 APSK. Secondly, the real and imaginary parts of the first stage symbol act as the two independent channels. Using convolution operation, the first stage symbol becomes the transmitted waveform value,
(1)XCAP(t)=Re{xup(t)}⊗fI(t)−Im{xup(t)}⊗fQ(t),  xup∈ℂ
where xup denotes the up-sampled first stage symbol. XCAP(t) is the transmitted signal waveform value. On the receiver side, the operation is similar. Firstly, there is a waveform-stage post-equalization to compensate for the linear/nonlinear loss in the waveform. Another convolution operation is performed by the same pair of filters to recover the two channels and restore the constellation of the first stage symbol.
(2){Re{xrup(t)}=XrCAP(t)⊗fI(t)Im{xrup(t)}=XrCAP(t)⊗fQ(t)
where XrCAP(t) is the received waveform value, xrup(t) is the recovered symbol before the down-sample process. There is a second stage post-equalization to further eliminate the inter-symbol-interference (ISI). The recovered symbol follows the same constellation mapping table to decode the symbol and compute the bit-error rate.

### 2.3. LMS/Volterra Post-Equalization 

LMS eliminates the linear error using the least mean square as the measure. It is used at both the waveform level and symbol level to pursue a thorough elimination. This process could be represented using the following equation [25]:(3)$$y\left(n\right)=\sum_{i=0}^{l-1}w_{i}\cdot x\left(n-i\right),\;\;\vec{w}=\left[w_{0},w_{1},\ldots w_{l-1}\right]=\underset{\vec{w}}{\arg\;\min}\sum_{j=1}^{N}{\left|y_{j}-\hat{y_{j}}\right|}^{2}$$

Volterra equalizer models the nonlinearity using the combination between all the terms ranged by the term order [25]:


(4)
$$\begin{array}{c} y_{\left(n\right)}=\sum_{k_{1}=0}^{N_{1}-1}w_{k_{1}}(n)\cdot x(n-k_{1})+\sum_{k_{1}=0}^{N_{2}-1}\sum_{k_{2}=k_{1}}^{N_{2}-1}w_{k_{1}k_{2}}(n)\cdot x\left(n-k_{1}\right)x\left(n-k_{2}\right)\\ +\sum_{k_{1}=0}^{N_{3}-1}\sum_{k_{2}=k_{1}}^{N_{3}-1}\sum_{k_{3}=k_{2}}^{N_{3}-1}w_{k_{1}k_{2}k_{3}}(n)\cdot x\left(n-k_{1}\right)x\left(n-k_{2}\right)x\left(n-k_{3}\right)+\ldots \\ +\sum_{k_{1}=0}^{N_{p}-1}\ldots \sum_{k_{p}=k_{p-1}}^{N_{p}-1}w_{k_{1}\ldots k_{p}}(n)\cdot x\left(n-k_{1}\right)\ldots x\left(n-k_{p}\right) \end{array}$$


In this experiment, we use 11 taps for the Volterra equalizer. The term number for the second-order terms is 121. If we use third-order Volterra, there are 1331 more terms. The gap in complexity forces us to choose the former one to provide a benchmark representing a conventional equalization solution that features low complexity and a quick training process. 

### 2.4. DNN Post-Equalization

Figure 1 briefly describes the structure of DNN. Its input is a feature vector constituted by the adjacent waveform samples (vector dimension equals the sample number). In addition, its output is the first stage symbol depending on the constellation mapping process. Since this mapping also includes offsetting nonlinear channel effect, it completes both the demodulation function and the post-equalization function as well.

All of the nodes are linked to the next adjacent layer. There is no linking backward or connections jumping over layers. One connection represents a multiplication and a bias operation. Mathematically, DNN performs projection and translation operations to the samples between and within vector spaces with different dimensions and finally separates samples into clusters on the 2D complex plane (the constellation plane).

When applying a nonlinear activation function (such as tanh), DNN is strengthened in fitting nonlinear functions. However, this also means that a slight disturbance to the input could result in a nonlinear response. Since DNN itself is a feedforward structure, it is sensitive to the linear error in the input. Although normalization is performed before feeding data into the DNN, there might be phase errors that result in a rotation in the output constellation. 

Meanwhile, the DNN is configured to adopt complex-valued weight and bias. Compared with using the two-dimensioned output to represent the real and imaginary parts of the output constellation, there is only one node at the output layer. One node means adjusting every weight connecting to the output layer using the same gradient [39]. However, when using two nodes at the output, there are two gradients corresponding to two nodes ($\frac{\partial E}{\partial y_{o,1}}$ and $\frac{\partial E}{\partial y_{o,2}}$). Weights connecting to one of the two nodes can only use either one of them. The separated adjustment could lead to independent fitting along the two dimensions and causes distortion. In the meantime, the loss function is not the mean square of the error, but the multiplication between the error and its conjugate to obtain a real-valued loss.

The hyperparameters are set in Table 1. The model for simulation is simpler than that for the experiment as it does not simulate the nonlinearity effect in a real channel.

### 2.5. Transfer Learning (TL) Strategy in NN-Based Post-Equalizer

According to the basic concept of homogeneous transfer learning in Figure 1, the key step is to divide the training set into the source domain and the task domain. The former contains the common knowledge of all the tasks and the latter should be the unique knowledge in solving a specific task. In this experiment, we assume that a model trained under a specific channel condition only has a slight difference to those prepared for other channel conditions: firstly, timing error causes a uniform linear error that appears as a ‘rotation’ in the constellation diagram; secondly, when the signal power increases, the nonlinearity gradually grows instead of randomly fluctuates. This process could be described as:(5)Xrotation(t)=r⋅(Φ[XTX(t)]+N(t)),  r,Xrotation,XTX,N∈ℂ
where XTX(t) is the transmitted waveform; Φ is the nonlinear function representing the nonlinear channel effect; N(t) is the noise; r is the rotational factor, Xrotation(t) is the received waveform.

Therefore, we choose the sample set obtained in moderate signal power as the source domain and the sets from other signal power in the task domain to further train the ‘stem model’ obtained from the training process in the source domain. The source domain consumes the most training epochs and the task domain could use relatively smaller training epochs by referring to the ‘stem model’ rather than beginning from a randomly initialized state, hence the computation cost is significantly reduced. 

### 2.6. AWGN Simulation 

To test the robustness of a single NN model to linear phase error, an AWGN channel is simulated using MATLAB code. The noise is kept at the same power level and hence the SNR increases as the signal power rises (proportional to the Vpp setting in this simulation). A random phase shift is added to the channel from 0 to 360 degrees. Such phase shift is constant for all the samples. It is a complex-valued rotation factor multiplied by the noisy receiving signal. As the NN model used in the simulation is linear, ignoring noise added, the rotation factor equals the angle of the ratio between the NN output and the label, represented by a complex value with a unity module. Using the mean value of such an angle can estimate the rotation and compensate for it by a pre-rotation before the NN. To estimate the angle only consumes less than 1000 symbols, less than 2% of the total symbol.
(6)Xrotation(t)=r⋅(XTX(t)+N(t)),  r,Xrotation,XTX,N∈ℂ
(7)Xpre−rotation(t)=Xrotation(t)r^,  Xpre−rotation,r^∈ℂ
(8)r^=1N ∑i=1nxr−y|xr−y|,r^, xr, y∈ℂwhere XTX(t) is the transmitted waveform; N(t) is the AWGN; r is the rotational factor; Xrotation(t) is the received waveform; Xcorrected(t) is the corrected waveform; xr is the NN prediction for the received symbol; y is the actual symbol (the label); and r^ is the estimated rotational factor.

### 2.7. Ring–Ring Diagram

To clearly demonstrate the nonlinearity effect, this paper uses a ring–ring diagram to show the dispersion of constellation points along the radial direction. The horizontal axis represents the original place of the symbol. The vertical axis means the position of the received symbol after the decision process. Figure 2 uses 64 APSK as an example. It has 4 rings and therefore it is a 4 by 4 matrix. The brightest diagonal line represents the symbols that remain at the same radial distance to the origin of the constellation. Supposing the communication system functions under the BER threshold, most of the constellation points should be distributed along the diagonal line. The upper violet area represents the symbols moving away from the origin and the lower area means the opposite case. If the channel demonstrates no nonlinearity, the violet area above and below the diagonal line should be approximately the same. However, due to the nonlinear distortion, the possibility for a constellation point to appear close to the origin than its original position is larger than in the opposite case. Furthermore, the violet area beneath the diagonal is significantly larger than that at the upper half of the matrix.

## 3. Results

### 3.1. AWGN Channel Simulation 

The simulation result is shown in Figure 3. The red and light bluish-gray lines represent the performance of a linear NN model in an Additive White Gaussian Noise (AWGN) channel when using a 64 APSK or 64 QAM modulation scheme. The BER performance quickly drops below the threshold at 3.8 × 10^−3^ BER for 7% hard-decision forward error correction (HD FEC). The lack of nonlinearity makes an increment in power always a beneficial factor to the BER. When the Vpp is above 650 mV, the BER is not measurable due to the limited length of bits transmitted. However, when a random phase shift is added to the channel, the constellation encounters a random rotation and the DNN used in this experiment is not capable to solve this problem. The BER increases dramatically. The orange and violet lines are located nearby BER = 0.5, indicating that the demodulation almost equals a random guess. However, since rotation is a linear and homogenous error to all the symbols, by multiplying a rotation factor before feeding the waveform samples into NN, the result (khaki and blue lines) is again as good as the ones without phase error. This simulation suggests that linear error would not cause a completely different result from the trained NN model, and only requires a small amount of information to be revised.

### 3.2. Underwater Experiment Result

#### 3.2.1. Exhaustive Training Strategy and Single Model Generalizability

Due to the complexity of VLC channel nonlinearity, this study chose the data collected in the experiment rather than simulation to obtain a signal with nonlinearity. Firstly, for each signal power, we trained both LMS/Volterra combination and DNN models to provide a benchmark and seeking for the model with the best BER performance as the ‘stem model’. Table 2 shows the BER performance of all the NN models trained using the waveform data from different Vpp configurations and compares it with LMS/Volterra combination. BER ratios are listed to provide a straightforward comparison.

It is obvious that DNN demonstrates a better BER performance, especially in the high Vpp area. The 64 QAM case is shown first. At 250 mV, the BER of DNN is 84.6% to that of the LMS/Volterra combination. At 550 mV, this ratio drops to 20.2%. Furthermore, this value reaches the lowest at 750 mV (8.7%). Only at the highest Vpp region does the performance of DNN degrade as the nonlinearity surges. The case for 64 APSK is also shown in Table 2 and generally follows the same pattern as the 64 QAM case. Its BER is 85.3% of that by the LMS/Volterra combination. This ratio decreases until Vpp = 650 mV (10.92%). Compared with 64 QAM, 64 APSK performs better in the low Vpp area but it is more vulnerable to nonlinearity in the high Vpp case. Among them, the waveform samples using 550 mV Vpp setting yield the best model trained in both cases. Therefore, these samples are regarded as the source domain and the models trained using them are recognized as the stem model. As its power level reaches a balance between SNR and nonlinearity severity. 

However, when applying the stem model to other Vpp settings, the performance decreases drastically. Figure 4 above shows that both linear and nonlinear loss appears. The constellation diagram demonstrates a clear rotation with respect to the expected one. This rotation suggests that the channel condition causes different phase shifts to the signal. 

Although the rotation is not as large as that in the simulation (random value from 0 to 360 degrees), it causes severe problems in low-power cases. In 64 QAM modulation, the BER in 250 mV increases by 310% compared with LMS/Volterra combination and 385% compared with the model trained using 250 mV Vpp samples. In 64 APSK cases, the BER performance is 18 times higher than LMS/Volterra combination and 21 times compared with the model trained using the 250 mV waveform samples. Meanwhile, the constellation also has a scaling error. It indicates that besides the difference in SNR, the linear error is the main factor limiting the generalizability. The nonlinearity in the high Vpp case also contributes to a different error. The ring-ring diagram shows that below the brightest diagonal line, there is a purple area showing that the demodulated symbols spread more into the area nearer to the origin. This phenomenon is more obvious when the Vpp is higher than 650 mV. Such nonlinearity grows significantly with the rising signal power. Due to the change in nonlinearity and the linear loss, the BER increases again. In the 750 mV 64 QAM case, the stem model results in BER 4 times and 56 times higher than that using LMS/Volterra combination and specifically trained NN model, respectively. In the 64 APSK case, the figures are 4.6 times and 48 times. The case at 850 mV shows that the demodulation actually failed in both 64 QAM and 64 APSK as the BER is higher than 0.5, worse than a random guess. This observation shows that the linear rotation and nonlinearity residual error are the main factors that limit the generalizability of the steam model. However, in most cases, the model successfully reproduces the grid shape of 64 QAM and 64 APSK, which means that the stem model completes the process of mapping waveform samples from a high dimension down to a 2D constellation. Even though the error causes a significant degradation in BER performance, through the constellation diagram and ring–ring diagram, the error is minor compared with a completely randomly initialized model. This gives hope to applying a transfer learning strategy.

#### 3.2.2. Transfer Learning Strategy

Figure 5 shows the gradually improved BER performance as the adaptive training iterations increase. The 64 QAM case is demonstrated first. When the extra training number is 2, the model has already demonstrated an astonishing performance. The working range reaches 300 mV, covering the Vpp from 400 mV to 700 mV. It is more than 4 times the range estimated when using the stem model only. This range is 58% larger than the working range of the LMS/Volterra combination. Compared with the result of an exhaustive training strategy, it achieves 64% of its working range. Further training increases the BER performance mainly in the high Vpp case. Tweny-five adaptive training epochs increase the working range to 420 mV, and successfully lowers the BER at 750 mV below the threshold. However, when 125 adaptive training epochs are applied, the working range only rises to 450 mV. Compared with the 25 epochs case, this increases the working range by 30 mV but consumes 100 training epochs. In the 64 APSK case, the situation is similar. With merely 2 epochs, the model set reaches 350 mV working range from 350 mV to 700 mV, about 40% more than that of the LMS/Volterra combination. The working range extends to 450 and 470 mV when 25 and 125 epochs are applied to tune the model parameter. It notes attention that when 25 epochs are applied, the result is almost the same as the exhaustive training strategy (450 mV to 470 mV). This improvement is more significant than that in the 64 QAM case. 

To demonstrate how the DSP strategy could benefit the system, the BER performance is converted into Q-value. The largest gap between Q-value appears at 750 mV. It shows that 2 adaptive training epochs in 64 QAM obtain a 1.5 dB gain compared with the LMS/Volterra combination. Twenty-five adaptive epochs contribute 2.6 dB, but 125 epochs only yield 2.8 dB gain. In the 64 APSK case, at the same Vpp, 2 epochs of adaptive training also bring a 1.5 dB gain. The gains at 25 epochs and 125 epochs are 2.5 dB and 2.8 dB, respectively. When the Vpp exceeds 750 mV, it requires more training epochs to approach the benchmark of an exhaustively trained model. 

### 3.3. Comparison between Two Strategies in Training Efficiency 

Figure 6 shows the comparison between the dynamic range and training efficiency measured by the division between the dynamic range and the training epoch numbers (including training the stem model). It demonstrates the same trends for both the 64 QAM and 64 APSK cases. Transfer learning generally outperforms exhaustive learning way. Transfer learning strategy could gain at most 5 times the training efficiency when adopting the listed training epochs (0.68/0.62 to 0.13). The peak of training efficiency is either 2 epochs or 25 epochs depending on the modulation scheme. The least efficient one with 125 epochs of adaptive training reaches nearly the same working range as the exhaustive training method from randomly initialized parameters, but still, it leads the latter by more than 150% in training efficiency, proving the effectiveness of the transfer learning strategy.

Figure 7 compares the difference in the loss function curve between the exhaustive training strategy and with stem model (ignoring the pre-training stage, which is close to the 650 mV group in the left diagram). It shows that all of the curves on the left consume approximately one order of magnitude of epoch number more to approach the same loss level. When training models from randomly initialized parameters, 250 mV waveform samples need about 150 epochs to make the loss function converge. However, in 650 mV and 850 mV cases, the loss function indicates that the model still does not find the minimum, leaving an under-fitted model. When starting from a ‘stem model’, the loss function converges quickly. In the 250 mV and 650 mV cases, at no.25 epoch, the loss functions already approach the result in exhaustively training strategy. The 850 mV case is significantly slower than the former two, and it consumes 125 epochs to reach the same level as randomly initialized models.

## 4. Discussion

In the simulation process, NN demonstrates its powerful mapping ability to convert high-dimension parallel waveform samples into correct constellation diagrams. The NN is even a linear model without adding a nonlinear activation function. In real cases, NN outperforms LMS/Volterra in the working range (more than doubled). It proves that DNN is a powerful tool to replace the conventional DSP process in demodulation.

However, due to the data-driven nature and feedforward structure, DNN does not have the ability to overcome even a minor disturbance to the input such as phase shift (a complex-valued rotation faction), which would result in a huge degradation in BER. Furthermore, experiment data also show that the generalizability of NN is challenged in real channel conditions. Considering that a nonlinear activation function is added to the DNN, the constellation is not only rotated but also distorted. Such distortion is a nonlinear response to the disturbance.

Adaptive training could quickly adjust the model to adapt to a lower power case. The linearity in this case is mild. However, a high Vpp case consumes a much large computation. The loss function indicates that even with a pre-trained ‘stem model’, the loss function in high Vpp training decrease very slowly compared with the middle or low Vpp case. This phenomenon means that the task domain in the high Vpp area shares less information with the source domain. Although, transfer learning successfully accelerates the learning process. The reason why 64 APSK is enhanced more than 64 QAM is that it utilizes high signal power more frequently, allocating more symbols to the peripheral area on the constellation diagram. Once the nonlinearity is alleviated, its potential in higher SNR compared with 64 QAM is able to demonstrate.

In this experiment, according to the range-efficiency diagram, either 2 epochs or 25 epochs of adaptive training is recommended. The 2 epoch setting is enough to deal with the low power signal and consumes the least computational resources. Although 25 epochs require a longer time to finish adaptive training, they could result in higher training efficiency. In comparison, 125 epochs are suggested not to apply as it brings marginal improvement disproportional to the computation consumption needed. If it is a must to extend the working range further, it is recommended to choose another sample set at a higher Vpp to enlarge the source domain but sacrifice the training time.

## 5. Conclusions

This experiment shows that transfer learning is capable of reducing the computational complexity in underwater VLC channels. Compared with LMS/Volterra (second order), it expands the working range by more than 50% but only requires 14.6% of the training epochs than the exhaustive training strategy. The epochs saved by transfer learning could be used to train more models for other channel conditions. In underwater communication, this feature will help the system to adapt to a rapidly changing condition of the channel environment or reduce the consumption of energy, making it more environmentally friendly, which shows that transfer learning would become a key technology in the 6G UVLC system. Nevertheless, this experiment only examines a naïve transfer learning approach to prove its viability, in future work it deserves more complex transfer learning methods to enhance its performance in large Vpp fluctuation by further reducing the training cost, especially in high Vpp regions.

## Figures and Tables

**Figure 1 sensors-22-09969-f001:**
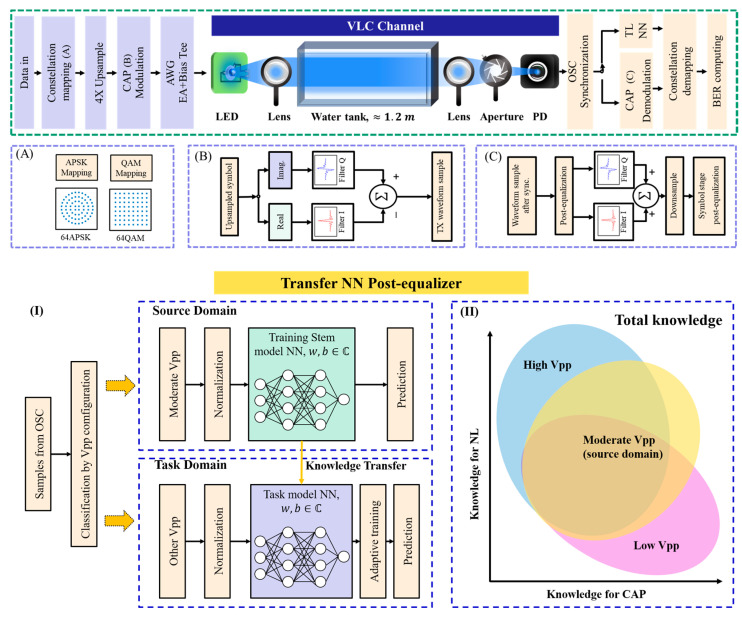
Upper part: block diagrams for the underwater VLC system. (**A**) The two choices for constellation mapping to generate the I and Q channels for CAP; (**B**) conventional modulation method for CAP; (**C**) conventional demodulation method for CAP. Lower part: transfer learning strategy proposed in this article. (**I**) The models are developed from a ‘stem model’ trained from the ‘source domain’. (**II**) A Venn diagram showing the knowledge distribution in data sets from different Vpp configurations. There is a trade-off between the knowledge for CAP demodulation and the nonlinear distortion. However, a moderate Vpp configuration would reach a balance between the two and, thus, is the best candidate for source domain.

**Figure 2 sensors-22-09969-f002:**
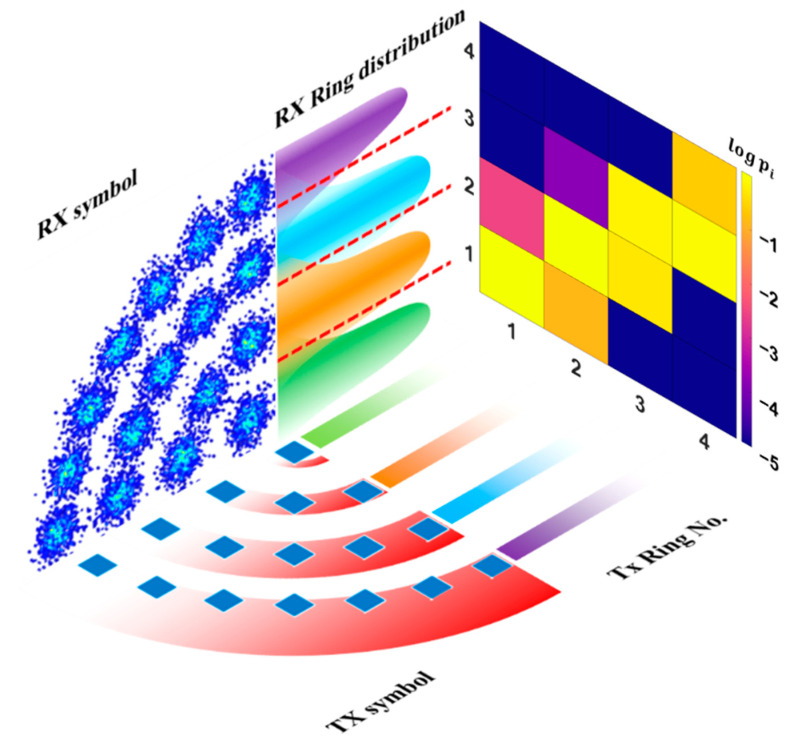
The illustration for ring–ring diagram. Its horizontal axis represents the ring position of the transmitted symbol, and the vertical axis represents the location of the demodulated symbol at the receiver. Noise and other channel effects spread the distribution along rings and deviate the symbols from the main diagonal line.

**Figure 3 sensors-22-09969-f003:**
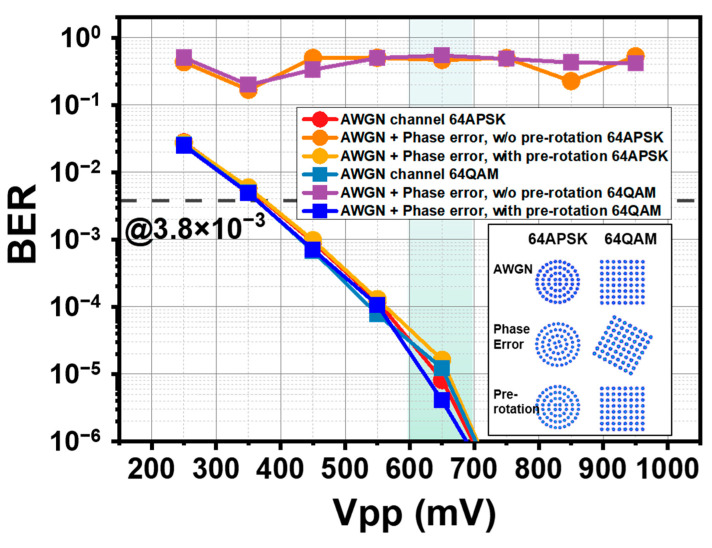
The simulation result for the NN demodulator. The subsets at the right bottom corner are corresponding to the samples shaded in the green bar. By sampling the phase error, the phase error of each symbol is demodulated, and the output is at the same level as AWGN case.

**Figure 4 sensors-22-09969-f004:**
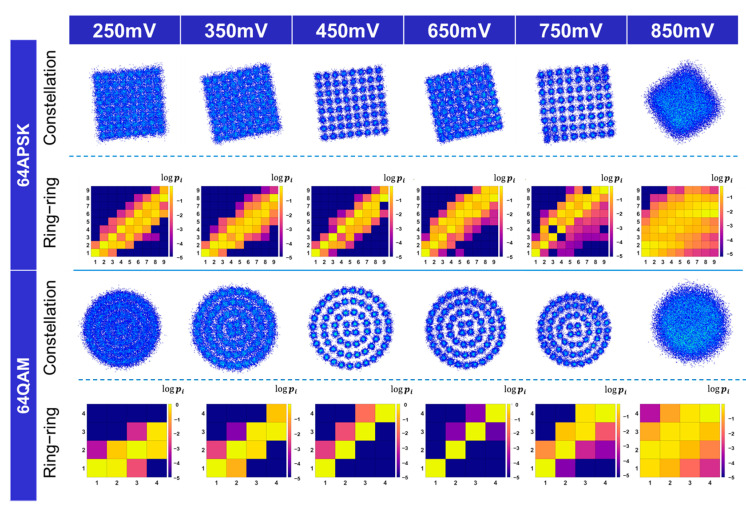
Constellation and ring–ring diagrams when waveform sets at other Vpp are applied to the model trained at 550 mV. The constellation demonstrates a clear phase shift and the ring–ring diagram shows the nonlinearity problem in high Vpp above 650 mV.

**Figure 5 sensors-22-09969-f005:**
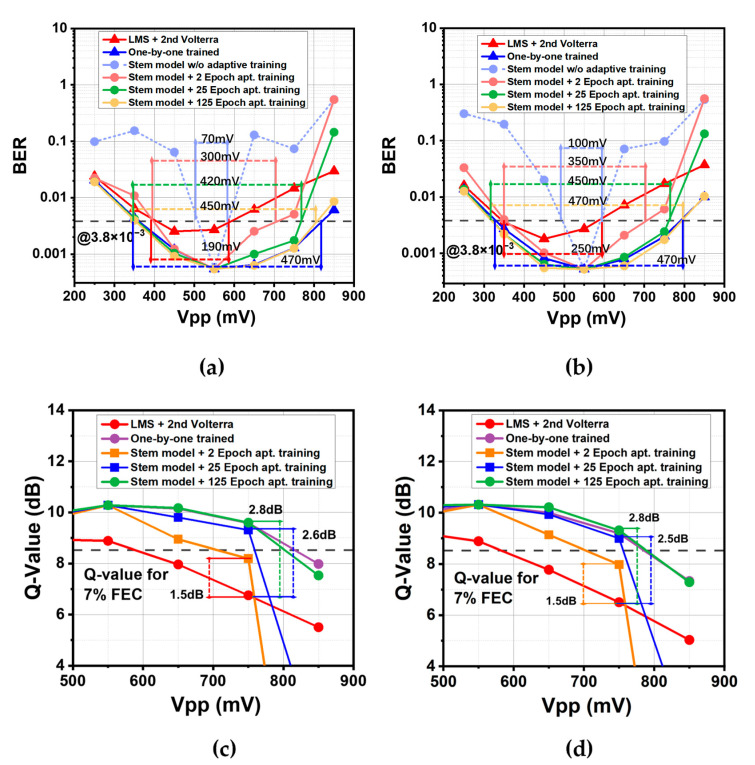
BER performance and Q-factor converted from BER. (**a**) BER curve for 64 QAM; (**b**) BER curve for 64 APSK; (**c**) Q-Value for 64 QAM; (**d**) Q-value for 64 APSK.

**Figure 6 sensors-22-09969-f006:**
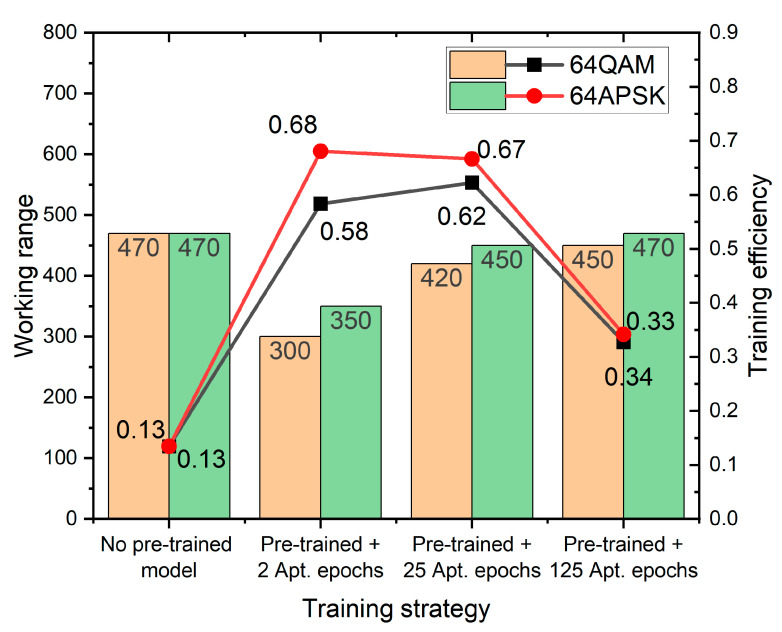
Working range versus training efficiency measured by the working range yield per training epoch. The maximum efficiency appears either at 2 adaptive epochs or 25 adaptive epochs. The 125 epochs setting only has a marginal advantage in the working range.

**Figure 7 sensors-22-09969-f007:**
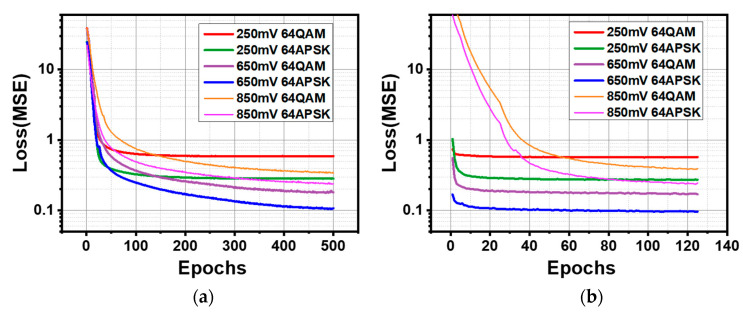
Loss curve in exhaustive (**a**) and transfer learning strategy (**b**). Transfer learning strategy has a significant advantage in converging speed. Higher signal power results in more complex nonlinearity and thus requires more training iterations.

**Table 1 sensors-22-09969-t001:** DNN parameters for three types of models used in the experiment.

Parameter	Simulation NN Model	Stem Model	Task Model
Input layer	60	60	60
First Hidden layer	16	16 (Tanh)	16 (Tanh)
Second Hidden layer	4	4	4
Output Layer	1	1	1
Training Epochs	100	500	* 2/25/125
Optimizer	Adam	Adam	Adam

* The training epochs for task model are set at 3 different scales to test their efficiency.

**Table 2 sensors-22-09969-t002:** BER performance between LMS/Volterra combination and DNN trained using exhaustive strategy. Models are trained from random initialization. The best BER indicates the training set that is an ideal source domain.

Constellation	Signal Vpp mV	BER LMS/Volterra (2nd Order)	BER DNN	BER Stem Model (550 mV)
64 QAM	250	2.41 × 10^2^	2.04 × 10^−2^	9.87 × 10^−2^
	350	6.42 × 10^−3^	4.38 × 10^−3^	1.54 × 10^−1^
	450	2.53 × 10^−3^	1.21 × 10^−3^	6.43 × 10^−2^
	550	2.71 × 10^−3^	5.46 × 10^−4^	5.46 × 10^−4^
	650	6.18 × 10^−3^	6.47 × 10^−4^	1.29 × 10^−1^
	750	1.48 × 10^−2^	1.29 × 10^−3^	7.37 × 10^−2^
64 APSK	250	1.60 × 10^−2^	1.37 × 10^−2^	3.03 × 10^−1^
	350	3.56 × 10^−3^	2.73 × 10^−3^	1.96 × 10^−1^
	450	1.81 × 10^−3^	8.19 × 10^−4^	2.00 × 10^−2^
	550	2.72 × 10^−3^	5.21 × 10^−4^	5.21 × 10^−4^
	650	7.20 × 10^−3^	7.86 × 10^−4^	7.13 × 10^−2^
	750	1.73 × 10^−2^	1.99 × 10^−3^	9.70 × 10^−2^

## Data Availability

Codes and data are not available online.

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
