# Peer review of "Transfer Learning Strategy in Neural Network Application for Underwater Visible Light Communication System"

_sensors, 2022, doi:10.3390/s22249969_

Round 1
Reviewer 1 Report
The research established that transfer learning strategy is especially suitable to solve the computational complexity problem in underwater VLC channel. The paper contributes transfer learning strategy in neural network scheme for underwater visible light communication, which enables keeping the outstanding advantage of neural network NN in VLC to alleviate nonlinear distortion while improving its versatility in underwater VLC channel. And the proposed scheme outperforms the state of the arts and can facilitate the application of NN to real communication system. The findings provide more insight in channel estimation/compensation process. However, they need to address the following concerns before the paper gets published.
Ø Major concerns
Ÿ The motivation of this research is not clear in the Abstract section.
Ÿ A clearer explanation that why you do this work should be given in the Introduction section.
Ÿ The structure of the article is not reasonable, such as there is too much content in Introduction section, which should present a current view of the problem we are exploring.
Ÿ The description of transfer learning strategies is not perfect and needs to be enriched.
Ÿ The proper nouns of DSP and OSC on the second page should be written in full.
Ÿ Insufficient description of the main techniques. Perhaps a new section to describe the use of the techniques would be just right.
Ÿ Discussion is not deep enough, or the Conclusions section needs more in it, as it's more of an afterthought. The authors are suggested to highlight important findings and include afterthought of this work.
Ø Minor concerns
Ÿ Short for “direct-modulation direct-detection method” in line 57 is inaccurate.
Ÿ Short for “additive gaussian white noise” in line 251 is inaccurate.
Ÿ The layout of Figure 1 is difficult to read and Figure 1 is a bit tedious, perhaps try to break it down.
Ÿ In line 152, the explanation of Figure 1 is repeated with the content above. Please check it out.
Ÿ Consider improving the clarity of the Figure1 &Figure 4.
Ÿ The Table 1 in your paper has a typographical error, the line number of the article was mistakenly placed in the table.
Ÿ The width of Table 2 should be consistent with that of body text, like Table 1.
Ÿ According to the MDPI Style Guide, e or E to mean “multiplied by the power of 10” is not allowed; please use the correct scientific notation for numbers, e.g., 3.7 × 105 (not 3.7e5 or 3.7E+5). Please check it out.
Ÿ The bottom of Equation 4 is incomplete and needs to be corrected.
Ÿ The Equation 4 label on the fifth page should be consistent with the format of the equation label above. It would be better to keep it centered.
Ÿ There are two consecutive commas in Equation 7 on page 7, so is there any information that you forgot to fill in?
Ÿ In line 355, the initial of the sentence should be capitalized.
Ÿ There are some Chinese translated statements.
Ÿ The authors may refer to some new studies in optical communication, such as
Yu, W.; Chen, F.; Xu, Z.; Zhang, Y.; Liu, A.X.; Zhang, C. Multi-Access Channel Based on Quantum Detection in Wireless Optical Communication. Entropy 2022 , 24, 1044. https://doi.org/ 10.3390/e24081044
Author Response
Please see the attachment, many thanks to your constructive and critical suggestions to improve our work.

Reviewer 2 Report
The manuscript present excellent research of underwater VLC including both experimental and analytical/simulation part to support proposed method of transfer learning for neural network. I suggest to accept it after major corrections:
1) authors mention through out whole paper nonlinearities, however those are not properly quantified. Can you please better describe them, parametrize in case of chose setup?
2) Can you compare influence of the inserted water tank to the case where is just only free space VLC link? Do you observe those above mentioned nonlinearities induced in the water case? Please explain more in detail
3) please revise formatting of references, some of them have missing authors etc.
Author Response

(The authors gave the same response as above.)

Round 2
Reviewer 2 Report
All my comments have been met and the manuscript revised accordingly so I suggest to accept the paper as it is